# Recurrent Inference Machines
# for Accelerated MRI Reconstruction

**Kai Lønning**
Informatics Institute
University of Amsterdam
`kai.lonning@gmail.com`

**Patrick Putzky**
AMLAB,
University of Amsterdam
`patrick.putzky@gmail.com`

**Matthan Caan**
Academic Medical Center,
Spinoza Centre for Neuroimaging
`m.w.a.caan@amc.nl`

**Max Welling**
AMLAB, University of Amsterdam,
Canadian Institute for Advanced Research (CIFAR)
`m.welling@uva.nl`

## Abstract

Accelerated MRI reconstruction is important for making MRI faster and thus applicable in a broader range of problem domains. Computational tools allow for high-resolution imaging without the need to perform time-consuming measurements. Most recently, deep learning approaches have been applied to this problem. However, none of these methods have been shown to transfer well across different measurement settings. We propose to use Recurrent Inference Machines as a framework for accelerated MRI, which allows us to leverage the power of deep learning without explicit domain knowledge. We show in experiments that the model can generalize well across different setups, while at the same time it outperforms another deep learning method and a compressed sensing approach.

## 1 Introduction

Magnetic Resonance Imaging (MRI) measures data in the space of net precession frequencies, known in MRI as k-space. Once enough samples in k-space are acquired to meet the Nyquist-criterion, the MR-image of tissue density can be computed through the inverse Fourier transform. However, there are physical constraints to the slew rate of the scanner's gradient system, putting a lower bound on the time it takes to fully sample k-space and produce an image. As such, aspirations to reducing MR scan times amount to acquiring k-space samples below the Nyquist-criterion and reconstructing the MR-image through a dealiasing algorithm. This amounts to solving what is known as an inverse problem. In this context, the forward model is a known process that describes the transformation taking the true image signal to the measured samples. What is not known, is the forward model's inverse transformation, taking the measured signal back to the true image, as this information is lost when k-space is sparsely sampled.

The set of possible MR-images is huge, even when restricted to a particular anatomical region, contrast mechanism or resolution. Also consider that each element spawns a large set of image corruptions, one for each permitted set of k-space sub-samples. As such, learning a direct mapping from each corruption back to the original signal, for all possible original signals, is a highly complex

1st Conference on Medical Imaging with Deep Learning (MIDL 2018), Amsterdam, The Netherlands.

problem, requiring an unfeasible number of parameters and a lot of data to train them. As such, part of the solution must help constrain the solution space. Well-established methods do this by careful design of features that exploit some known property inherent to MRI. In this work, we apply Recurrent Inference Machines (RIM) for accelerated MRI reconstruction, which were first proposed in [7] as general inverse problem solvers. They constrain the solution space by learning an iterative process, where step-wise reassessments of the maximum a posteriori estimate result in incremental updates that infer the inverse transform of the forward model.

Apart from breaking a complex problem into multiple sub-problems, we conjecture that this iterative "meta-learning"-approach prevents the model from overfitting on the image statistics of the dataset, by shifting focus toward learning the inversion procedure itself. This should result in a larger degree of invariance with respect to changes in the specific imaging settings. In this paper, we show that RIMs can accurately and efficiently reconstruct sparsely sampled MR-images at varying acceleration factors, and that their solution is robust against perturbations in sub-sampling points, image resolution levels, and to some extent the underlying anatomy being imaged, making them suitable candidates for distribution across different MR-acquisition set-ups.

## 2   Background and related work

We start by introducing the forward model of the inverse problem of accelerated MRI reconstruction. Let $\mathbf{x} \in \mathbb{C}^n$ be the true image signal and $\mathbf{y} \in \mathbb{C}^m$, $m \ll n$, be the set of sparsely sampled frequency signals measured by the scanner in k-space. The sampled measurements can then be described in terms of the true image,

$$\mathbf{y} = P\mathscr{F}\mathbf{x} + \mathbf{n}. \tag{1}$$

Here, the MR-image is projected onto its frequency domain through the Fourier transform $\mathscr{F}$, followed by a sub-sampling mask $P$, which discards some fraction of values in k-space, thereby facilitating the acceleration in scan times. Further, samples are assumed to be subjected to additive, normally distributed noise $\mathbf{n} \sim \mathcal{N}\left(0, I\sigma^2\right) + i\mathcal{N}\left(0, I\sigma^2\right)$ stemming from various measurement errors accumulated by the scanner. While MR-scanners have multiple receiver coils, which each produce partial images with different spatial sensitivity, we assume for simplicity in this work that there is only one receiver coil with homogeneous spatial sensitivity.

The goal in accelerated MRI reconstruction is to find an inverse transform of the forward model in (1), thereby mapping incomplete measurements $\mathbf{y}$ to a high resolution image $\mathbf{x}$. Usually, this is dealt with by solving the regularized optimization problem

$$\underset{\mathbf{x}}{\operatorname{argmin}} \left\{ d\left(\mathbf{y}, P\mathscr{F}\mathbf{x}\right) + \lambda R\left(\mathbf{x}\right) \right\}, \tag{2}$$

where $d$ evaluates the data consistency between the reconstruction and measurements, and $R$ is a regularizer preventing overfitting, with regularization factor $\lambda$. Whereas the data-consistency term in (2) follows explicitly from the forward model in (1), the regularizer $R$, representing the prior distribution over MR-images, is more a matter of model design.

A solution to (2) is typically found through an iterative process. One such example is the well-established method known as Compressed Sensing (CS) [6], which uses a transformation known to compress a particular sub-set of MR-images, combined with the $\ell_1$-norm, which yields sparse solutions when used as a regularizer. Relative to deep learning, this shallow form of feature extraction leads to a highly efficient algorithm, however, depending on the network architecture, learned features tend to win in terms of accuracy. As such, the use of deep neural networks is increasing, also in medical imaging [5].

In recently proposed deep learning-based solutions that model gradient descent on (2), the prior is learned by specifically designated convolutional layers [14][4]. Another paper reinforces data consistency in separate layers, interspersed between longer sequences of convolutional layers [10]. Other deep learning proposals move away from (2) altogether. For example, in [8], the Generative Adversarial Network's (GAN) overall success is repurposed for the task of removing aliasing artifacts in a non-iterative fashion, whereas in [3], they deploy the U-net architecture known from image segmentation tasks [9], in an attempt to learn a direct mapping from $\mathbf{y}$ to $\mathbf{x}$.

As in [14] and [4], the RIM also learns an iterative scheme based on (2). However, instead of using separate parameters for each iteration, the RIM has a recurrent architecture, sharing weights across

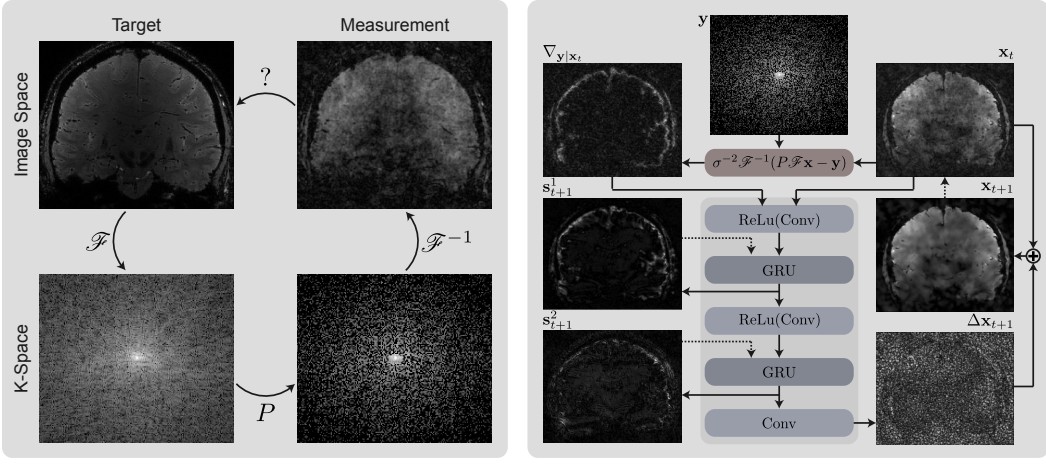

**(a)** Measurement Process          **(b)** RIM architecture

**Figure 1: (a)** The goal in MRI is to retrieve a high resolution image (top left). Meaurements are done in k-space (bottom left) which is related to image space through a Fourier transform. In order to accelerate the measurement process, the k-space is sub-sampled (bottom right). Imaging the sub-sampled k-space meaurement will lead to an aliased image (top right). The goal of this work is to find a function that maps from an incomplete k-space (bottom right) to a high resolution image (top left). **(b)** The RIM update function $h_\phi$ as used in this work. All images show magnitudes scaled individually. Magnitudes of intermediate internal states $s^1$ and $s^2$ were averaged over features. Bold lines depict connections within a single timestep, whereas dotted lines represent recurrent connections that pass information to the next time step.

iterations, while utilizing internal states to distinguish them. Following Recurrent Neural Network (RNN) conventions, we shall hereby refer to these iterations as time-steps. We do not use the same input, but this approach bears resemblance to the model described in [1], where RNNs are trained to learn gradient descent schemes by using the gradient of the objective function as the network's input for each time-step.

## 3 Recurrent Inference Machines

In the context of accelerated MRI reconstruction, the gradient of the log-likelihood, corresponding to the data-consistency term $d$ in (2), is given by $\nabla_{\mathbf{y}|\mathbf{x}} := \sigma^{-2} \mathscr{F}^{-1} (P \mathscr{F} \mathbf{x} - \mathbf{y})$. As for the problem of evaluating the gradient of the log-prior distribution, corresponding to $R$ in (2), this is solved by passing the current state $\mathbf{x}_t$ as an input to the network, such that any function that would implicitly approximate the log-prior gradient can be evaluated at $\mathbf{x}_t$. The RIM's iterative scheme can then be described by the following update equations:

$$
\begin{aligned}
\mathbf{s}_0 &= \mathbf{0}, & \mathbf{x}_0 &= \mathscr{F}^{-1}\mathbf{y}, \\
\mathbf{s}_{t+1} &= g_\phi\left(\nabla_{\mathbf{y}|\mathbf{x}_t}, \mathbf{x}_t, \mathbf{s}_t\right), & \mathbf{x}_{t+1} &= \mathbf{x}_t + h_\phi\left(\nabla_{\mathbf{y}|\mathbf{x}_t}, \mathbf{x}_t, \mathbf{s}_{t+1}\right),
\end{aligned}
\tag{3}
$$

for $0 \leq t < T$. $h_\phi$ denotes the RIM's neural network, parametrized by $\phi$, such that each pass through $h_\phi$ produces the next incremental update $\Delta \mathbf{x}_t$ in the RIM's iterative scheme. $g_\phi$ is simply the part of the network responsible for producing the next internal state $\mathbf{s}_{t+1}$, which the RIM needs in order to keep track of time-steps and modify its behaviour based on the progression of the inference procedure.

In this work, the update function $h_\phi$ was implemented using a sequence of alternating convolutional layers and gated recurrent unit (GRU) cells, where the first two convolutional layers are followed by ReLU activation functions before the feature maps are passed to the GRUs. The GRU cells work as described in [2], and are assigned the task of maintaining the internal state, meaning that in practice there are two states $\mathbf{s} = \left\{\mathbf{s}^1, \mathbf{s}^2\right\}$ represented by $\mathbf{s}$ in (3). Figure 1b illustrates the way in which these layers were assembled. The current estimate $\mathbf{x}_t$ is simply concatenated with the log-likelihood gradient $\nabla_{\mathbf{y}|\mathbf{x}_t}$ along the channel dimension to produce the input of the first convolutional layer, resulting in 4 input channels due to the complex components also being given separate channels. This

first layer is implemented with a kernel size of $5 \times 5$, whereas the next two convolutions have kernel sizes $3 \times 3$. All convolutions are padded to retain the same image size through-out the network. We will take $F$ to mean the number of features in the GRU cells' states and the number of feature maps produced by the convolutional layers. This hyper-parameter is kept the same through-out all internal layers, before the final convolutional layer outputs the complex-valued image update $\Delta \mathbf{x}_t$. Note that the GRU cells' weights are shared across image pixels, but differ across the feature maps produced by the convolutional layers, allowing the network to process images of any given size.

We use the mean square error (MSE) as a loss function, where the estimate $\mathbf{x}_t$ is evaluated against the true image $\mathbf{x}$ for each time-step. The total loss to minimize is then given by the average per-pixel MSE,

$$L\left(\mathbf{x}_T\right) = \frac{1}{nT} \sum_{t=1}^{T} \|\mathbf{x}_t - \mathbf{x}\|_2^2 , \tag{4}$$

where, as before, $n$ is the total number of image pixels, and $T$ is the total number of time steps during training.

## 4 Experiments

### 4.1 Dataset

On a $7.0\,\mathrm{T}$ Philips Achieva scanner (Achieva, Philips Healthcare, Cleveland, USA) equipped with a 32-channel Nova head coil, 3D $T_2^*$-weighted multi-echo FLASH data was acquired with an isotropic resolution of $0.7\,\mathrm{mm}^3$ and Field-of-View (FOV) $224 \times 224 \times 126\,\mathrm{mm}^3$, matrix size $320 \times 180$, 320 slices with transversal slice encoding direction, 6 echoes with echo times (TEs) ranging from $3\,\mathrm{ms}$ to $21\,\mathrm{ms}$, repetition time (TR) $23.4\,\mathrm{ms}$, flip angle $12°$, and second order image-based B0 shimming. The data were fully sampled with an elliptical shutter, such that the total scanning time was $22.5\,\mathrm{min}$.

On a $3.0\,\mathrm{T}$ Philips Ingenia scanner (Philips Healthcare, Best, The Netherlands) equipped with a 32-channel head coil, $T_1$-weighted three-dimensional (3D) magnetization prepared rapid gradient echo (MPRAGE) data were acquired with an isotropic resolution of $1.0\,\mathrm{mm}^3$ and FOV $256 \times 240\,\mathrm{mm}^2$, matrix size $256 \times 240$, 225 slices with sagittal slice encoding direction, TFE factor 150, shot interval $2500\,\mathrm{ms}$, inversion delay $900\,\mathrm{ms}$, flip angle $9°$, and first order shimming. The data were fully sampled with an elliptical shutter, such that the total scanning time was $10.8\,\mathrm{min}$.

On both scanners, raw data was exported and stored for offline reconstruction experiments. Coil sensitivities were estimated from the data using auto-calibration, after which data of different coil elements was combined [11]. The data was then normalized with respect to the maximum magnitude value in the image domain.

12 healthy subjects were included, from whom written informed consent (under an institutionally approved protocol) was obtained beforehand.

### 4.2 Acceleration method

Previous papers on deep learning methods for accelerated MRI reconstruction train separate models for each acceleration factor they evaluate on [4, 8, 14, 3, 10]. Whether this is known to be necessary through empirical testing or due to a working assumption is unclear. Either way, our hypothesis is that the RIM is capable of dealing with a range of acceleration factors. All models in this paper were trained on acceleration factors that were randomly sampled from the uniform distribution $U\left(1.5, 5.2\right)$. Sub-sampling patterns were then generated using a Gaussian distribution, thereby favoring low frequencies that contain more information about the general shape of the image content, while also creating incoherent noise due to randomness. We thereby adhere to the requirement of incoherent noise, as established in compressed sensing [6]. Furthermore, the signals emitted near the origin in k-space were always fully sampled within an ellipse with half-axes set to 2% of the image axes.

Model selection was done by testing on acceleration factors 2x, 3x, 4x and 5x, using the best average of three constant sub-sampling patterns within each acceleration category. The selected models were then evaluated on the hold-out datasets for final comparisons, using 10 sub-sampling patterns within each acceleration factor. Examples of acceleration patterns used for evaluation can be seen in Fig. 2.

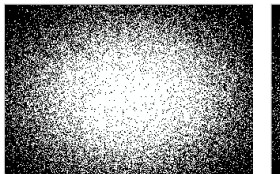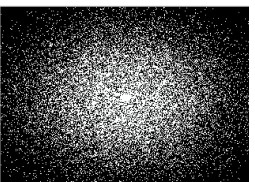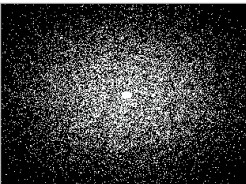

**Figure 2:** Examples of sub-sampling masks used for testing. Acceleration factors are, from left to right: 2x, 3x, 4x and 5x.

### 4.3 Research focus

Concerns referring to the data-driven black box-nature of deep learning algorithms are sometimes raised as a point of preference toward more traditional approaches. Even if they may come with performance costs, non-learned models always extract the intended features and thus do as prescribed regardless of the data they receive. Furthermore, MR-images are acquired at a large variety of settings, ie. different contrast mechanisms and levels of signal-to-noise and resolution. A reconstruction model should be able to generalize well across different settings typically used in the lab or clinic, preferably even without having encountered the particular setting during training.

With this in mind, we wish to illustrate the RIM's ability to generalize to images acquired under different conditions than those contained in the training set. We believe that, due to its internal states and by including the forward model during inference, the RIM is capable of separating features related to image statistics from the features that have to do with the inversion procedure itself, and therefore generalizes well to new types of data.

In this work, we train one RIM on the $3\,\mathrm{T}\ 1.0\,\mathrm{mm}$-dataset using slices along the transverse plane (RIM 3T), and another on the $7\,\mathrm{T}\ 0.7\,\mathrm{mm}$-dataset in the coronal plane (RIM 7T), after which we cross-evaluate their performance. Unless otherwise mentioned, the models were trained on randomly cropped patches of size $200 \times 200$, which were randomly rotated, flipped and mirrored for data augmentation purposes. The training sets consisted of 10 subjects, whereas 1 subject was used for model selection and a final subject for final evaluation. We will use the Structural Similarity index (SSIM) [13] as a metric for quantifying the reconstruction quality.

For comparing against other reconstruction methods, we will use CS and the U-net architecture [3]. CS is chosen because it is currently being used for online reconstruction in many scanners today, whereas the U-net is chosen for its ease of implementation and training, and because it is already a well-known architecture for per-pixel tasks in deep learning.

For CS, we use the BART toolbox described in [12]. We use the $\ell_1$-norm of the wavelet transform as a regularizer, with regularization factor 0.05 and 40 iterations. For the U-net, we used nearly the same architecture as described in [3]. However, we achieved better results using max-pooling instead of average pooling when downscaling the image. We also trained the network using the ADAM optimizer, and no post-processing step was used. Finally, a big difference with our implementation is that we compare on the same acceleration method as described in 4.2, whereas in [3] they use a one-dimensional periodic sub-sampling scheme which remains constant through-out training. We found that, unlike the RIM, the U-net could only learn to reconstruct the magnitude of an image, and would fail when attempting to learn the real and imaginary components in order to obtain a phase reconstruction as well.

We begin the experiments by investigating the effect of varying a couple of the RIM's hyper-parameters, namely the number of time-steps $T$ and the number of features $F$.

## 5 Results and discussion

### 5.1 Hyper-parameters

The two plots in Fig. 3 show the SSIM scores on the model validation set of the final reconstructions for RIMs trained on different values of $T$ and $F$. These models were trained on smaller patches of size $30 \times 30$.

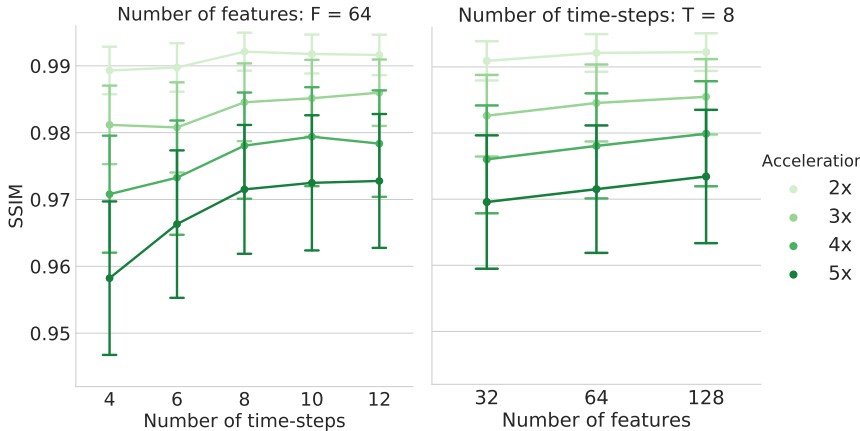

**Figure 3:** The plots show SSIM-values for the final reconstruction of RIM models trained on different hyper-parameters. Left figure: The effect of varying the number of features. Right figure: The effect of varying the number of time-steps.

We first look at the effect of $T$, where we wish to find the optimal number of updates needed for the RIM to learn its iterative scheme until convergence. Setting the number of features to $F = 64$, we trained RIMs over a total of 4, 6, 8, 10 and 12 time-steps. The fewer the number of time-steps, the fewer the number of training iterations are necessary for the model to learn. We also found that, the fewer the time-steps a model is trained on, the greater the reconstruction improvement is between time-steps. In other words, the 4th estimate of a model trained on 4 time-steps will be better than the 4th estimate produced by a model trained on 6 time-steps. However, the 6th estimate of a model trained on the 6 time-steps will be even better. Moreover, learning becomes unstable when training on a low number of time-steps. Especially when training on 4 time-steps we observed diverging loss-curves as it approached a local minimum, requiring the use of gradient clipping and several restarts for successful training. Training on 6 time-steps was also problematic, whereas at 8 time-steps this problem had subsided.

The higher the acceleration factor, the more there is to be gained by increasing the number of time-steps. However, the improvement from training on 8 to 10 time-steps is marginal, whereas going from 10 to 12 time-steps even decreases the SSIM score for certain acceleration levels. We suggest that this decrease is best interpreted as noise due to differing mini-batch sizes and human error involved in determining when a model had converged sufficiently to stop training. At this point, the benefit from spending more time on training larger models becomes negligible, and there is also a question of inference speed, hence we proceed with models trained on 8 time-steps going forward.

Next, we ask what there is to be gained from increasing the number of RIM features. Keeping $T = 8$, we train three models setting the number of features to $F = 32, 64, 128$. Unlike when increasing the number of time-steps, we do not observe a marked drop in improvement when doubling the number of features. This would imply that we should increase the number of features further. However, we consider the time it takes to reconstruct an image to be too long at 128 features, and so we make a compromise and select 64 features going forward.

### 5.2 Generalization and overall comparisons against Compressed Sensing and the U-net

Fig. 4 illustrates the SSIM scores of the different models evaluated on all acceleration levels. The two plots show the difference when the models are evaluated on the two datasets described in 4.1. As seen, the RIMs are capable of producing nearly the same results on both types of data, regardless of the type of data they were trained on, even beating CS when evaluated on the unseen dataset. The U-nets show a greater tendency to overfit on the type of data used for training, as seen by the drop in performance on the unseen dataset. Also note that the whisker plots, corresponding to the standard deviation in SSIM scores, are narrower for the two RIM models than for CS and the U-net, indicating that the RIM is more robust against variations in sub-sampling patterns or input images.

Qualitative results can be seen for samples from the 3T and 7T test sets in Fig. 6 and Fig. 5, respectively. Extended figures with acceleration factors 2, 3, and 4 can be found in A.

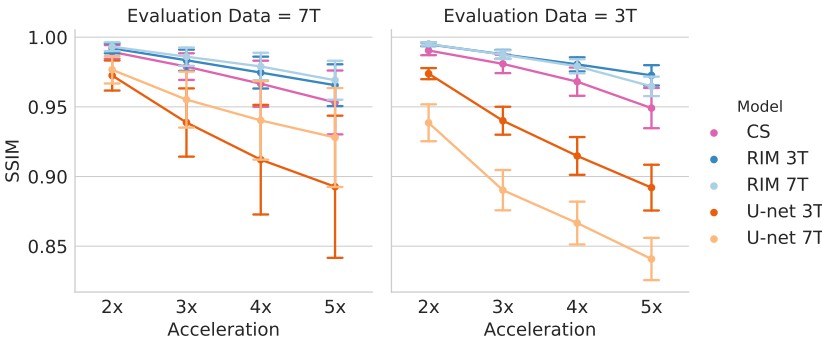

**Figure 4:** SSIM of final reconstructions of RIM models that were trained on datasets from the $3\,\mathrm{T}$-scanner at $1\,\mathrm{mm}$ resolution, and from the $7\,\mathrm{T}$-scanner at $0.7\,\mathrm{mm}$ resolution. Both models generalize well to the resolution they were not trained on, even out-performing compressed sensing.

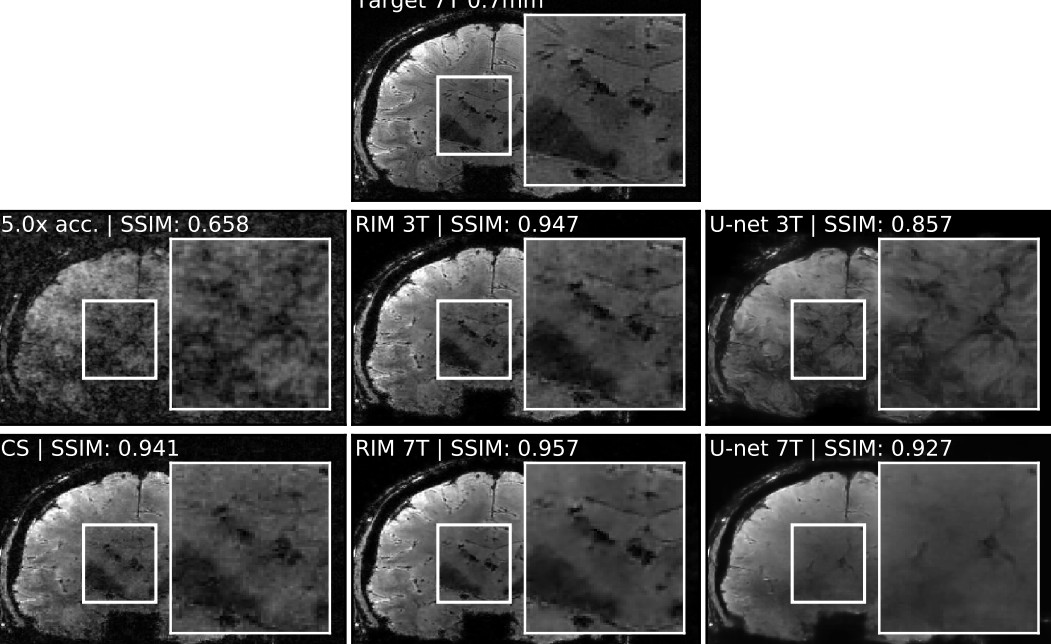

**Figure 5:** Reconstructions of a sample image from the 7T test set at acceleration factor 5x. The two RIMs and U-nets were trained on 3T and 7T datasets, respectively.

As seen in the figures, RIMs produce less noisy reconstructions with more detail preserved when compared to CS. The reconstruction quality is also highly consistent, whereas images produced by CS vary with respect to the image and sub-sampling pattern used. The same holds when comparing RIMs with U-nets, but to a larger extent. Our results indicate that U-nets tend to overfit on data type and the specific pattern used for sub-sampling, and are likely to have suffered under the randomized acceleration method used in this work during training. What the RIM can extract from its current reconstruction state, log-likelihood gradient, and hidden states, the U-net must make up for in terms of model parameters, which, at 1,328,833 versus 94,336 parameters, will more likely lead to overfitting.

As for reconstruction times, the results shown for the RIM in this paper take $236\,\mathrm{ms}$, although this time can be reduced by lowering the number of features or time-steps without sacrificing too much in terms of reconstruction quality, as indicated by Fig. 3. For instance, with $F = 32$, the reconstruction time becomes $126\,\mathrm{ms}$. At $40\,\mathrm{ms}$, the U-net is 6 times faster than the RIM, which may make it preferable for low acceleration factors at high resolution levels. These aforementioned times are measured on a single GPU. Unfortunately, we only have CPU time for CS, which is $1203\,\mathrm{ms}$, however, we expect that CS might be faster than the RIM when applied on the GPU.

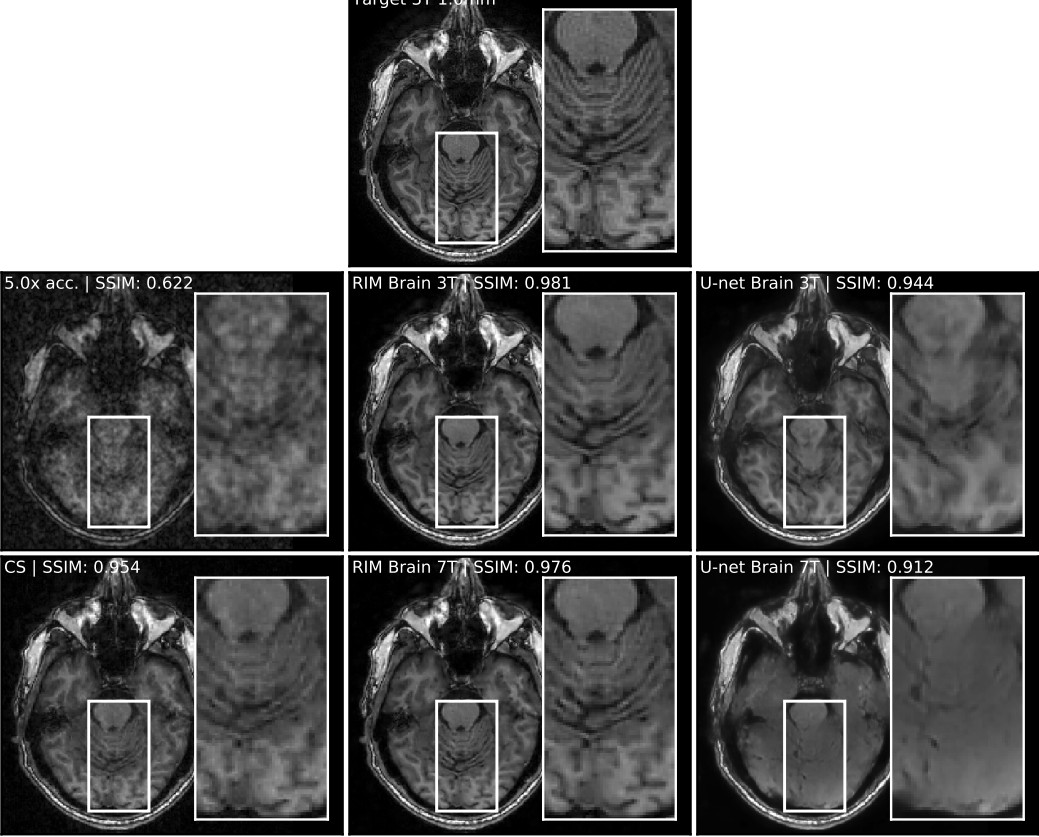

**Figure 6:** Reconstructions of a sample image from the 3T test set at acceleration factor 5x. The two RIMs and U-nets were trained on 3T and 7T datasets, respectively.

# 6    Conclusion

We have shown that RIMs are capable of accelerated MRI reconstruction, thereby producing reconstructions of higher quality than the U-net and Compressed Sensing. Reconstructions were less sensitive to varying input images and perturbations in the sub-sampling pattern used for data acquisition. Further, we have shown that RIMs generalize well to datasets with unseen levels of resolution and signal-to-noise ratios, and even to data recorded from a different angle than what was trained on. This emphasises the RIMs ability to perform well across the various measurement scenarios that a present in clinical practice. Furture work will show how well this method can generalize over other measurement scenarios such as different types of anatomical data and different numbers of recording coils.

**Acknowledgments**

We kindly thank dr. F.M. Vos and dr. G.A.M. Arkesteijn for their support in data acquisition.

Kai Lønning and Max Welling are supported by the Canadian Institute for Advanced Research (CIFAR).

Patrick Putzky is supported by the Netherlands Organisation for Scienetific Research (NWO) and the Netherlands Institute for Radio Astronomy (ASTRON) through the big bang, big data grant.

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

# A    Reconstructions on acceleration factors 2x, 3x, and 4x

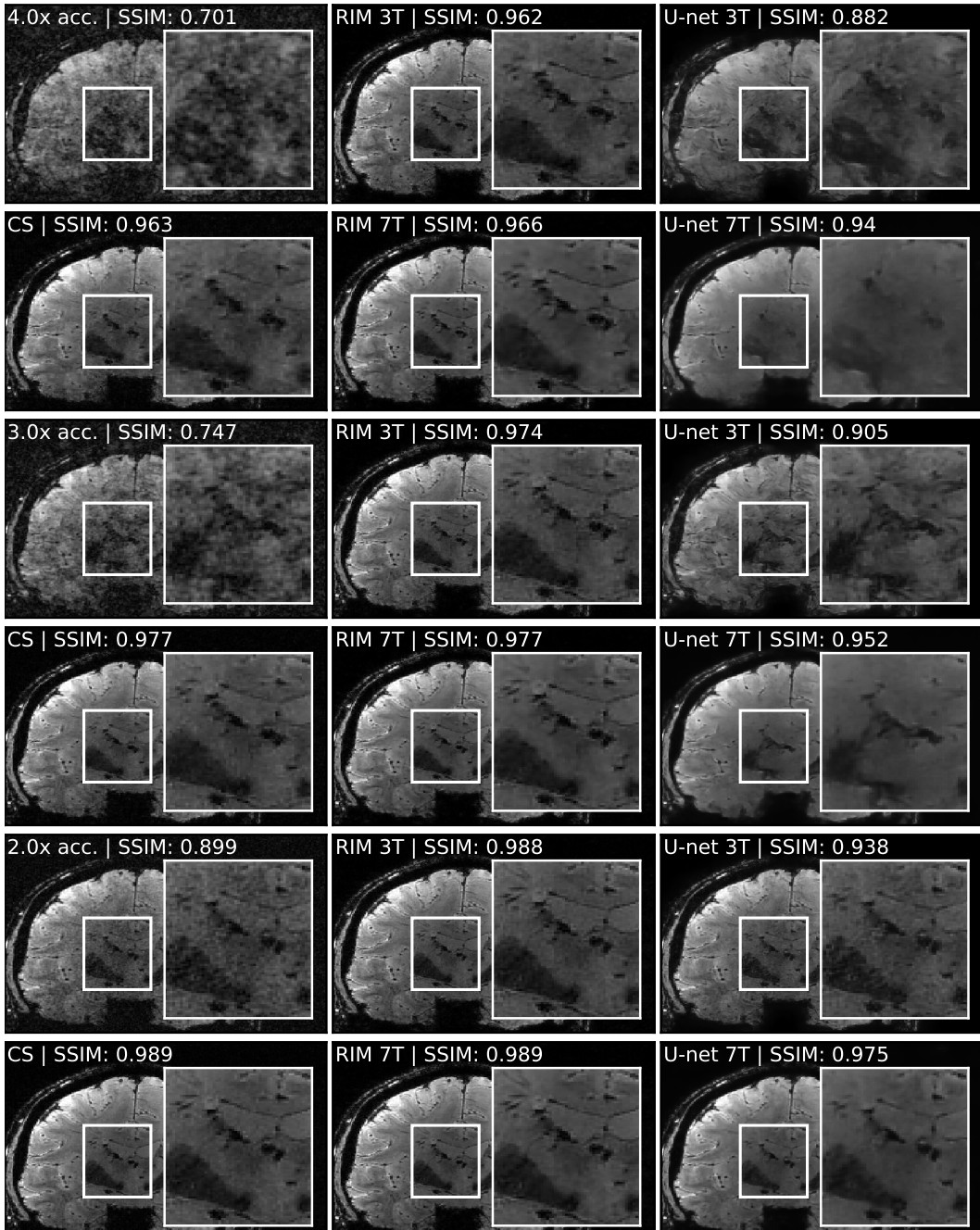

**Figure 7:** Final reconstructions of an image from the 7T test set, at acceleration factors 2x, 3x, 4x and 5x. The two RIMs and U-nets were trained on 3T and 7T datasets.

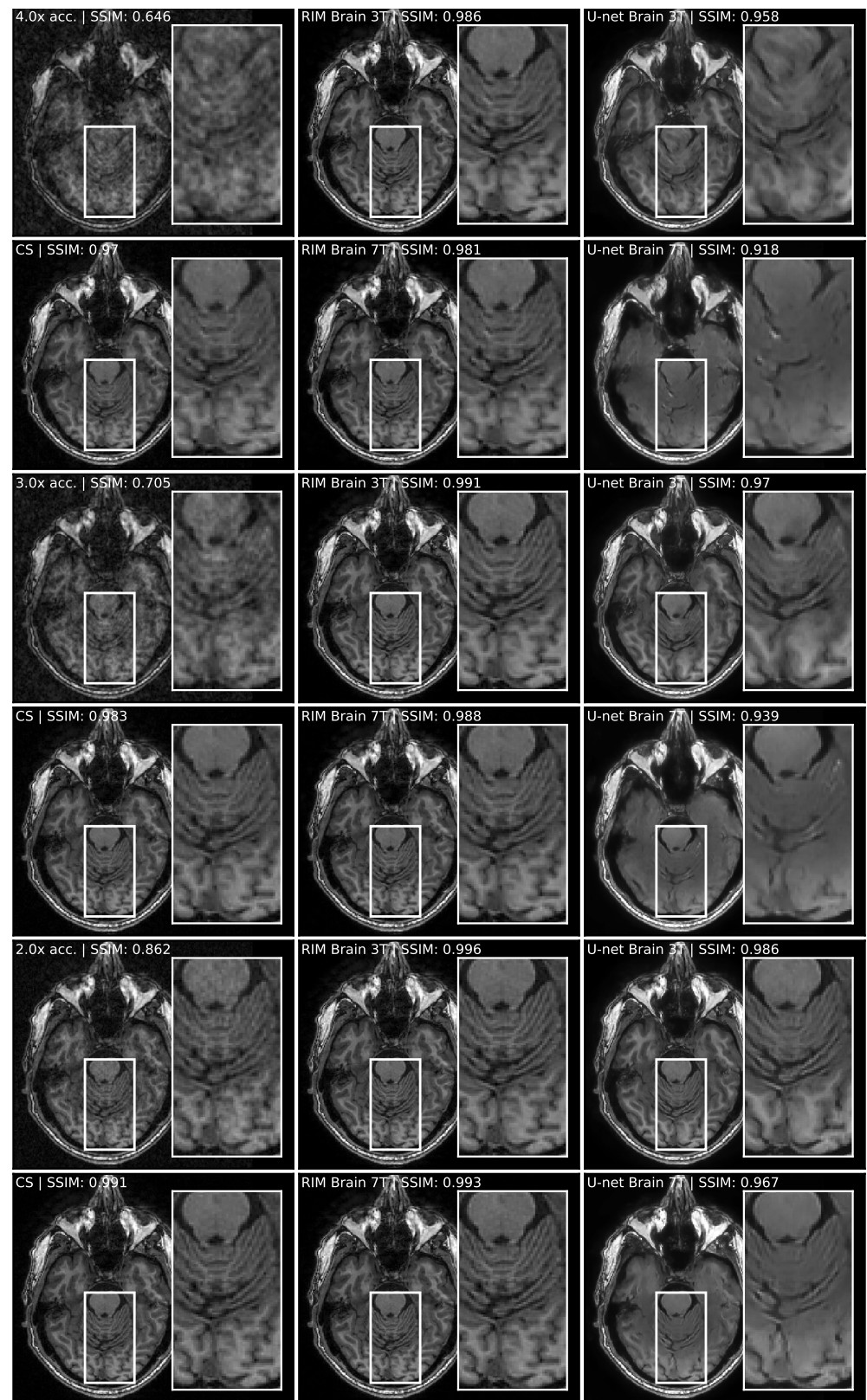

**Figure 8:** Final reconstructions of an image from the 3T test set, at acceleration factors 2x, 3x, 4x and 5x. The two RIMs and U-nets were trained on 3T and 7T datasets.

