# OpenReview forum: "Recurrent Inference Machines for Accelerated MRI Reconstruction"
_MIDL.amsterdam/2018/Conference — MIDL 2018 Oral_

### Review · AnonReviewer2 · 2018-05-01
**Recommended**

**Rating:** 4
**Confidence:** 2

**Review:**

This work clearly demonstrates that their novel method, RIM, is outperforming in MRI reconstruction compared to conventional methods. Overall, the manuscript is well-written and the experiments are well-organized. Highly recommended.
Minor comments:
1) it would be great if it includes some sort of discussion from a clinical point of view for the usefulness of the proposed method over the conventional one. For example, how small features in brain MRI (e.g., aneurysm) can be reconstructed using the proposed method / conventional method?
2) Please fix typos. For example, '... high resultion image x', which is probably 'resolution'
3) Please make references consistent to each other. Also, some references have missing information, e.g., no journal information in [4].

**Special Issue:**

Yes

---

### Review · AnonReviewer1 · 2018-05-09
**Interesting work but authors need to highlight the novel aspects**

**Rating:** 2
**Confidence:** 2

**Review:**

This paper presents a recurrent inference machines (RIM) framework to accelerate the reconstruction of magnetic resonance imaging (MRI). The RIM model is a combination of alternating convolutional layers and gated recurrent units (GRU). A ReLU activation function is used after the convolution layers. This study is conducted on datasets from two scanners (7.0 T & 3.0 T), where the model was trained on randomly cropped patches of size 200 x 200 after performing data augmentation. The reconstruction quality is estimated by using structural similarity index. The paper also shows the reconstruction of MR images at different acceleration factors (2x, 3x, 4x, 5x). Overall the RIM outperformed the U-net model and a compressed sensing approach. The paper is well written and the problem is well explained.

Below are some major and minor comments, which I would like the authors to comment on
1- The proposed method is based on a previously published work (RIM model) and as such, this reviewer found it difficult to find any significant novel contribution(s) in this paper as compared to the RIM model [7]. I would encourage the authors to highlight the novel aspects of their work and if there are any modifications or task-specific information incorporated within previously published RIM model.
2- In the comparative analysis, it is not clear which compressed sensing algorithm was used for comparison and there is no reference available for the reader. It would be helpful to add the appropriate citation and briefly explain the reason for selecting that particular compressed sensing algorithm.
3- The reconstruction time for U-net architecture is 6x faster than RIM model. It would be interesting to briefly explain the reason (U-net is fully convolutional could be one of the reason) or provide an estimate for the number of trainable parameters involved on both models.
4- The motivation behind selecting the RIM model to solve this problem can be improved further.
5- In Fig 1b, it would be more informative to highlight which part is h(theta) and g(theta) and the overall flow of the diagram can be improved.
6- Is there any particular reason for using 30x30 patch size for hyper-parameters selection instead of 200x200?


**Special Issue:**

No

---

### Review · AnonReviewer3 · 2018-05-17

**Rating:** 4
**Confidence:** 2

**Review:**

This paper suggests the usage of recurrent inference machines (RIM) for MRI reconstruction fro  data subsampled in K-space for accelerated imaging. 12 healthy subjects were included in the study and randomised sampling in K-space with factor 2-5 are used. The RIM shows a clear improved reconstruction over CS methods.

**Special Issue:**

No

---

### Comment · ~Bram_van_Ginneken1 · 2018-05-18
**Selection for longlist for special issue Medical Image Analysis**

Dear authors,

Congratulations on your acceptance to MIDL! We have selected your paper on the longlist for the Medical Image Analysis Special Issue. Please read this page:
https://midl.amsterdam/special-issue-in-medical-image-analysis/
Please answer the three questions that are listed on that page about your interest in submitting to the special issue, potential overlap with other publications, and related publications.

You can post your answer here directly below on openreview.net, or mail me directly at bram.vanginneken@radboudumc.nl.

Best regards, Bram

---

### Decision · Program_Chairs · 2018-05-15
**Paper74 Acceptance Decision**

Oral